# Consumer acceptance of personalised nutrition: The role of ambivalent feelings and eating context

**Machiel J. Reinders** *, **Emily P. Bouwman, Jos van den Puttelaar, Muriel C. D. Verain**

Wageningen Economic Research, Wageningen University & Research, Wageningen, The Netherlands

* machiel.reinders@wur.nl

**Data Availability Statement:** All relevant data are within the manuscript and its Supporting Information files.

## Abstract

Providing dietary suggestions based on an individual's nutritional needs may contribute to the prevention of non-communicable dietary related diseases. Consumer acceptance is crucial for the success of these personalised nutrition services. The current study aims to build on previous studies by exploring whether ambivalent feelings and contextual factors could help to further explain consumers' usage intentions regarding personalised nutrition services. An online administered survey was conducted in December 2016 with a final sample of 797 participants in the Netherlands. Different models were tested and compared by means of structural equation modelling. The final model indicated that the result of weighing personalisation benefits and privacy risks (called the risk-benefit calculus) is positively related to the intention to use personalised nutrition advice, suggesting a more positive intention when more benefits than risks are perceived. Additionally, the model suggests that more ambivalent feelings are related to a lower intention to use personalised nutrition advice. Finally, we found that the more the eating context is perceived as a barrier to use personalised nutrition advice, the more ambivalent feelings are perceived. In conclusion, the current study suggests the additional value of ambivalent feelings as an affective construct, and eating context as a possible barrier in predicting consumers' intention to use personalised nutrition advice. This implies that personalised nutrition services may need to address affective concerns and consider an individual's eating context.

## 1. Introduction

The incidence of non-communicable dietary related diseases such as type-2 diabetes, and cardiovascular disease is growing [1]. In this respect, Fallaize et al. reported the insufficiency of a 'one-size-fits-all' approach to dietary guidance [2]. Instead, dietary interventions tailored to individuals' personal nutritional needs by what have been termed 'personalised nutrition services' show the potential to improve healthy dietary patterns and public health [3,4]. There are increasing indications that personalised dietary advice is more effective compared to nutrition information that is provided to consumers in a generic manner [3,5–8]. Personalisation has the potential to make the provided nutrition information more personally relevant [9], and

**Funding:** The research reported is supported by the Dutch Personalised Nutrition and Health program (www.personalisednutritionandhealth. com) funded by the Dutch Top Sector Agri & Food (TKI-AF-15262). The funder had no role in study design, data collection and analysis, decision to publish, or preparation of the manuscript.

**Competing interests:** The authors have declared that no competing interests exist.

may in turn lead to greater compliance than generic dietary advice. In this respect, personalised nutrition services can be defined as the delivery of personalised dietary advice based on information regarding an individual's characteristics such as lifestyle (i.e. dietary intake and personal preferences), phenotype (i.e., blood (bio)marker analysis, anthropometric measurements), and/ or genotype (i.e., DNA profile) [10,11]. Enabling momentum for personalised nutrition services can be realised by developments in information technology, such as health parameter monitoring, big data handling, and the high prevalence rate of smartphone ownership (up to 40% worldwide and 87% in Dutch households [12,13]. Moreover, technological do-it-yourself tools for health measurement and monitoring (e.g. smart wearables) are rapidly becoming available, and consumer interest in these online health tools, such as smartphone apps, is growing [14]. These developments and technologies enable personal data collection and nutrition delivery services, and practitioners are beginning to use these tools to personalise nutritional advice [15,16].

Despite this potential, consumer reticence to use personalised nutrition services could compromise the possible benefits resulting from personalised nutrition [17]. It would therefore be advantageous to understand how consumers come to accept personalised nutrition. In this respect, several recent studies have shown that, in addition to the motivation necessary to engage in healthy behaviour, the intention to use personalised nutrition services is determined by a decision-making process that weighs the risks and benefits of the personalised nutrition approach [18–21]. More specifically, these studies on consumer acceptance of personalised nutrition indicate that consumers' intention to adopt personalised nutrition services increases with greater perceived benefits, and fewer perceived risks associated with personalised nutrition [22,19]. For example, Berezowska et al. suggested that consumers feel that disclosing their privacy information is too great a risk compared to the benefits offered by personalised nutrition services [22]. These researchers capture this 'benefits and risks' trade-off in an overall information disclosure valuation, called privacy calculus. This privacy calculus is operationalised as a mediator between perceived personalisation benefits and perceived privacy risks on the one hand and the adoption intention of personalised nutrition services on the other. Note that the perceived benefits of personalised nutrition services are intrinsically associated with the *concept* of personalised nutrition—that is, the anticipated individual health and fitness benefits received from personalised dietary recommendations. On the other hand, perceived risks reflect consumers' concerns associated with the *delivery system* of personalised nutrition advice—that is, concerns about the handling of privacy-sensitive information [21].

However, consumers' adoption intention regarding personalised nutrition services may not be the result solely of cognitive considerations of benefits and risks. Several studies in other contexts suggest that affective factors also play an important role in explaining consumers' adoption intention of new (technology-based) services [23,24]. In this respect, several studies point out the role of mixed feelings in predicting the adoption intention of new technologies and services [25,26]. Aversive or mixed feelings are grounded in the experience of internal conflict (i.e., ambivalence) and in turn negatively affect intentions [27]. Ambivalence, which refers to holding both positive and negative evaluations simultaneously, may result from conflicting evaluations of benefits and risks [28,29]. Generally, studies have found that it is the affective nature of ambivalence (in this study labelled as ambivalent feelings) that drives the effects of ambivalence on behavioural intentions [30,31]. Van Trijp and van Kleef explain ambivalence in the domain of food product innovation with two fundamental human tendencies: neophobia (i.e., the fear of novelty) and neophilia (i.e., the urge towards embracing novelty) [32]. As a result, humans alternate between approach (neophilia) and avoidance (neophobia) for new products. As personalised nutrition may simultaneously stimulate both the urge towards a better health and fear of data misuse, ambivalent feelings among consumers

may very well be the result [30]. As such, we propose ambivalent feelings as an additional explanatory construct, in addition to the cognitive risk-benefit consideration, to explain consumers' adoption intention based on benefits and risks.

Moreover, a limited number of studies indicate that eating context poses a barrier to the adoption of personalised nutrition. For instance, eating outside the home can be a potential barrier in adhering to healthy eating intentions [33]. Additionally, the competing dietary requirements of other family members may hinder adherence to healthy eating intentions [34–36]. Personalised nutrition advice is usually adjusted to one individual and therefore might not be suitable in situations where a meal is shared with multiple individuals, such as family members. Furthermore, Stewart-Knox et al. suggested that consumers perceive difficulty in adhering to personalised nutrition advice in social situations, such as eating at the homes of friends or acquaintances [21]. As such, the eating context may act as an additional barrier in the uptake of personalised nutrition services. We therefore propose one's perceived eating context as another construct in explaining consumers' intentions for adopting personalised nutrition services.

To summarise, the aim of this study is to build on previous research in the area of personalised nutrition by exploring whether ambivalent feelings and perceived eating context barriers further explain consumers' intentions to adopt personalised nutrition services. Additionally, our work illuminates how these aspects may help to further explain the link between perceived risks and benefits (as captured by the concept of risk-benefit calculus) and intention to adopt personalised nutrition services.

## 2. Methods

### 2.1 Sample and procedure

An online survey was conducted in December 2016 in the Netherlands with a total of 1,000 participants. The survey was administered by MSI-ACI Europe BV, a professional market research company. Participants were sampled from its consumer panels and approached by email to fill out a self-administered online questionnaire. At the start of the questionnaire potential participants received information about the study in which it was explained that participation is on a voluntary basis and that data are treated anonymously. The purpose of the research was also explained. Agreeing with this information before starting the questionnaire constituted informed consent.

To ensure a sample that was nationally representative, participants were quota-sampled based on demographic characteristics: gender, age, highest level of completed education and income. Some participants were removed from the final sample. First of all, four completed questionnaires were ineligible due to missing values or incorrect answers. To avoid bias in the results that may be caused by including people who currently use personalised nutrition advice—and who may see its benefits and risks differently, based on their experiences—an additional 40 participants, who have experience with personalised nutrition services, were omitted from the analyses. The target group of our study are people who are not experienced with personalised nutrition services, as we want to gain more insight in the acceptance potential of personalised nutrition services. Finally, to ensure that participants understood the provided definition of personalised nutrition, we asked them to rate on a scale ranging from 1 (not at all) to 7 (totally) whether they had a clear picture of personalised nutrition advice. Only those who provided a score equal to or higher than the midpoint of the scale (4 or higher), were eligible for the study. This resulted in a further 159 participants being removed from the sample. Taken together, the final sample consisted of 797 individuals. This sample was composed of 391 males and 406 females, with a mean age of 47.5 years and an age range from 18 to

75 years. Overall, 23.7% of the participants had a low education level by the standards of Dutch education, 38.8% had a moderate education level, and 37.5% had a high education level. Generally, in the Dutch education system, low education indicates only primary school and lower secondary education, moderate education indicates higher secondary education and vocational education, and high education includes those with university, postgraduate degrees, and professional education. For more details about education levels, refer to the questionnaires (both original and translated versions) in S1 and S2 Data.

Further demographic characteristics of the study sample are shown in Table 1.

The questionnaire included several parts. At the beginning of the questionnaire, a definition of personalised nutrition was provided (see text box below).

After answering a question as to whether they understood the meaning of personalised nutrition advice based on the provided definition, participants were required to assess the measurement scales of the constructs that were used in the model described in this study (Note that measurement scales of some additional constructs were included in the questionnaire. Since they were beyond the scope of this study, these results are not reported here. Data regarding these constructs were collected for another purpose). The final part of the questionnaire comprised inquiries about the psychographic and demographic background characteristics of the participants.

Before data collection started, researchers pre-tested several versions of the questionnaire. Pre-tests of early versions of the questionnaire focussed on whether the questions and items were understandable and well-formulated. Different online versions of the questionnaire were then pre-tested to detect any final errors and to test the length of time that it took to complete the questionnaire. The study was approved by the Social Sciences Ethics Committee of Wageningen University & Research in the Netherlands.

**Table 1. Sample characteristics (N = 797).**

| | Percentage | | Percentage |
|---|---|---|---|
| **Gender** | | **Work status** | |
| Male | 49.1% | Self-employed (with employees) | 2.8% |
| Female | 50.9% | Self-employed (without employees) | 7.2% |
| **Age** | *Mean* = 47.5 years (*SD* = 15.9); *Range*: 18–75 years | Fulltime employee | 25.7% |
| 18–25 | 11.7% | Part-time employee | 14.6% |
| 26–35 | 15.1% | Temporary/ seasonal work | 0.3% |
| 36–45 | 17.7% | Fulltime housewife/-husband | 8.5% |
| 46–55 | 21.1% | Fulltime student | 7.0% |
| 56–65 | 19.1% | Unemployed | 10.4% |
| > 65 | 15.4% | Retired | 17.9% |
| **Education** | | Other | 5.6% |
| Low | 23.7% | **Number of people in household** | |
| Medium | 38.8% | Single | 27.0% |
| High | 37.5% | 2 persons | 40.4% |
| **Net monthly income** | | 3 persons | 12.8% |
| Low (up to € 1.500) | 25.4% | 4 persons | 14.4% |
| Medium (€ 1.500 to € 3.000) | 44.2% | 5 persons | 4.3% |
| High (more than € 3.000) | 30.4% | 6 or more persons | 1.1% |
| I don't know/ won't say | 20.8% | | |

> ### Box. Definition of personalised nutrition as provided to the participants
>
> *Personalised nutrition advice is advice that is completely tailored to you as a person. Advice can be personalised in different ways: (1) personal preferences (e.g., what food you like and don't like), (2) self-formulated goals/ -wishes (e.g., lowering cholesterol; enhancing fruit consumption), (3) eating habits, (4) health status (e.g., overweight, blood pressure and cholesterol) or (5) DNA profile.*
>
> *Information technology developments ride on this personalisation trend with 'health trackers', like smart watches, pedometers and smartphone apps, which help you to gain insight in your own behaviour or health.*
>
> *An example:*
>
> - *Generic nutrition advice: general nutrition guidelines prescribe that you eat whole grain products on a daily basis.*
>
> - *Personalised nutrition advice: based on your personal profile and preferences, we advise you to eat yoghurt with oatmeal instead of whole grain bread.*

## 2.2 Measures

A number of well-established, validated scales were used to measure the various constructs. The original questionnaire items were translated into Dutch and back-translated as appropriate. All measurement scale items were arranged on seven-point answering scales, and, except for *Risk-Benefit Calculus* and *Ambivalent Feelings*, which use semantic differentials, were operationalised as Likert scales ranging from 1 (strongly disagree) to 7 (strongly agree), with 4 being the (neutral) midpoint of the scale. The measurement items can be found in Table 2. Furthermore, we refer the reader to the S1 and S2 Data.

*Intention to Use Personalised Nutrition Advice* was measured by means of three items based on Poínhos et al. [19]. *Risk-Benefit Calculus* was measured with one item based on the semantic differential scale used by Berezowska et al. [22]. Participants answered the question "Do you think using personalised nutrition advice will provide greater benefits than risks, or greater risks than benefits?" on a scale ranging from 1 (greater risks) to 7 (greater benefits). *Ambivalent Feelings* were measured with three items on semantic differentials based on Berndsen and Van der Pligt [37]. Participants indicated whether they would feel conflict, uneasiness and mixed feelings should they use personalised nutrition advice. *Personalisation Benefit* was measured with three items which state whether personalised nutrition is more accurate, relevant and beneficial for someone's health needs and is based on Berezowska et al. [22] and Xu et al. [38]. *Privacy Risk* was measured with three items to determine whether personalised nutrition presents a risk to one's privacy and is based on Berezowska et al. [22] and Xu et al. [38]. Furthermore, we measured *Eating Context Barrier* by means of five items, based on Stewart-Knox et al. [33]. For different eating situations, participants were required to indicate to what extent each context would prevent them from using personalised nutrition advice.

All constructs were averaged across their scale items to create a composite construct score. All constructs and their items, means, standard deviations, factor loadings, Cronbach's alphas, and average variance extracted (AVE) are shown in Table 2. Factor loadings, Cronbach's alphas, and AVE were used to test the reliability and validity of the measurement scales. As a

**Table 2. Measurement items, means, factor loadings, and reliability and validity checks (N = 797).**

| Measures and items | $M^*$ | SD | λ | CR | AVE |
|---|---|---|---|---|---|
| **1. Intention to Use Personalised Nutrition Advice** | 3.86 | 1.46 | | .90 | .76 |
| I intend to use personalised nutrition advice. | 3.64 | 1.59 | .95 | | |
| I would consider using personalised nutrition advice. | 4.38 | 1.65 | .80 | | |
| I am definitely going to use personalised nutrition advice. | 3.55 | 1.56 | .87 | | |
| **2. Risk-Benefit Calculus**<br>All things considered, do you think using personalised nutrition advice will offer greater benefits than risks, or greater risks than benefits? | 4.88 | 1.23 | .92 | | .84 |
| **3. Ambivalent Feelings**<br>When I would use personalised nutrition advice . . . | 3.50 | 1.51 | | .90 | .75 |
| . . .I feel no conflict at all/I feel maximum conflict | 3.55 | 1.65 | .85 | | |
| . . .I feel no uneasiness at all/I feel maximum uneasiness | 3.33 | 1.69 | .85 | | |
| . . .I have no mixed feelings/ I have strong mixed feelings | 3.63 | 1.62 | .89 | | |
| **4. Personalisation Benefit**<br>Compared to general nutrition advice, personalised nutrition advice offers me advice that . . . | 5.16 | 1.13 | | .92 | .79 |
| . . .Is more accurately tailored to my health needs | 5.19 | 1.23 | .88 | | |
| . . .Is more relevant for my health | 5.09 | 1.25 | .89 | | |
| . . .Is more beneficial for my health | 5.21 | 1.20 | .89 | | |
| **5. Privacy Risk**<br>I think that using Personalised Nutrition advice . . . | 3.47 | 1.53 | | .92 | .79 |
| . . .Involves many privacy-related risks | 3.71 | 1.63 | .82 | | |
| . . .Is a threat to my privacy | 3.33 | 1.67 | .92 | | |
| . . .Creates a high risk for the loss of my privacy | 3.36 | 1.67 | .92 | | |
| **6. Eating Context Barrier**<br>What would prevent you from using personalised nutrition advice? | 4.43 | 1.23 | | .83 | .60 |
| Providing different foods for family members. | 4.10 | 1.74 | .52 | | |
| Difficulties in maintaining healthy eating habits when eating out in restaurants. | 4.73 | 1.51 | .79 | | |
| Difficulties in maintaining healthy eating habits when eating at other people's houses. | 4.81 | 1.49 | .82 | | |
| Difficulties in maintaining diet when travelling. | 4.59 | 1.56 | .79 | | |
| Difficulties maintaining diet when at work. | 3.94 | 1.70 | .62 | | |

$M$ = mean (* constructs measured on scales 1 to 7); SD = standard deviation; λ = standardized factor loading; CR = composite reliability (Cronbach's alpha);

AVE = average variance extracted

measure of scale (composite) reliability, we used Cronbach's alpha to test the internal consistency of the measurement items of the constructs, that is, how closely related a set of items are as a group. Table 2 shows that Cronbach's alpha was higher than .80 for all constructs, indicating good reliability. In addition, to measure the validity of the constructs, we looked at convergent and discriminant validity. Convergent validity refers to the degree to which two measurement items that theoretically should be related, are factually related. Except for looking at the inter-item correlations between the individual items of a scale (which should be greater than .50), criteria for convergent validity require the AVE greater than 0.5, standardized factor loading of all items not less than 0.5, and composite reliability not less than 0.7 [39]. As can be seen in Table 2, the variables that were used have high factor loadings (> .50), high composite reliabilities (Cronbach's alphas >0.7) and AVE greater than 0.50, which indicates convergent validity. Additionally, discriminant validity tests whether measurements that are not supposed to be related are actually unrelated. Discriminant validity can be tested by checking whether the square root of the AVE for each construct is greater than its correlations with the other constructs [40]. When looking at Tables 2 and 3 we can conclude that this was indeed the case for our constructs, indicating that all constructs in our model possess good discriminant validity.

**Table 3. Descriptive statistics and inter-correlations among study variables (N = 797).**

| | $M^*$ | SD | 1. | 2. | 3. | 4. | 5. | 6. |
|---|---|---|---|---|---|---|---|---|
| 1. Intention to Use Personalised Nutrition Advice | 3.86 | 1.46 | - - | | | | | |
| 2. Risk-Benefit Calculus | 4.88 | 1.23 | .43** | - - | | | | |
| 3. Ambivalent Feelings | 3.50 | 1.51 | -.31* | -.39** | - - | | | |
| 4. Personalisation Benefit | 5.16 | 1.13 | .52** | .55** | -.36** | - - | | |
| 5. Privacy Risk | 3.47 | 1.53 | -.13** | -.34** | .46** | -.22** | - - | |
| 6. Eating Context Barrier | 4.43 | 1.23 | .02 | -.08* | .31** | .01 | .25** | - - |

$M$ = mean (* constructs measured on scales from 1 to 7), SD = standard deviation;

* $p < .05$;

** $p < .01$.

## 2.3 Data analyses

The researchers obtained an anonymised data set from the market research agency. Descriptive statistics were computed using means (*M*) and standard deviations (*SD*) for the variables. Associations between variables were tested using Pearson correlation coefficients. To test the measurement model, the internal consistency of the multiple-item scales was investigated by performing a first-order confirmatory factor analysis (CFA) on all items. The CFA was performed with maximum likelihood estimation in the R package lavaan [41]. Note that in running the CFA in R, and to deal with the fact that *Risk-Benefit Calculus* was a single-item measure, the error variance (i.e., 0.24) of this measure was specified as the variable's covariance with itself. Measurement error can be incorporated into a single indicator by fixing its unstandardized error to some nonzero value, calculated on the basis of the measure's sample variance estimate and known psychometric information (e.g., internal consistency estimate). The reliability was derived from prior research that generalizes well to the sample for which the current analysis is being undertaken [42].

After testing the fit of the measurement constructs to the underlying data by means of CFA, different models were tested using structural equation modelling with maximum likelihood estimation in R. More specifically, three steps were distinguished to test the relations between the latent constructs. First, a model was estimated with *Personalisation Benefit* and *Privacy Risk* as independent variables, *Risk-Benefit Calculus* as mediator and *Intention to Use Personalised Nutrition Advice* as the dependent variable. Second, a model was estimated with *Ambivalent Feelings* as an additional mediator. Third, the model was extended further with *Eating Context Barrier* as an additional independent variable.

Model fit for both the measurement model and the structural models was assessed based on a number of fit statistics. The criterion for acceptance of the Chi-square estimate with degrees of freedom ($\chi^2$/df) varies from less than 2 to less than 5 [43–45]. Comparative Fit Indices (CFI) and Tucker-Lewis Indices (TLI) of at least .90 indicate a satisfactory model fit [42]. A Root Mean Square Error (RMSEA) of .07 or lower and a Standardized Root Mean Square Residual (SRMR) below .08 indicate satisfactory model fit [46,47].

# 3. Results

## 3.1 Descriptive results

Table 3 shows the means, standard deviations and zero-order correlations among all variables included in the models. The construct with the highest mean score was found to be in *Personalisation Benefit* (*M* = 5.16). This score, which is well above the midpoint of the scale (which is a

score of 4), suggests that participants perceive benefits to personalisation. The mean score on *Privacy Risk* ($M$ = 3.47) is just below the scale midpoint, which suggests that participants perceive moderate risks associated with receiving personalised nutrition advice. These scores also suggest that participants perceive greater benefits than risks, which is also reflected in the mean score on *Risk-Benefit Calculus* ($M$ = 4.88). This score, which is above the scale midpoint, suggests a positive balance towards more perceived benefits than perceived risks. The mean score on *Ambivalent Feelings* ($M$ = 3.50) is just below the midpoint of the scale, suggesting that participants have moderately ambivalent feelings about using personalised nutrition advice. In contrast, the mean score on *Eating Context Barrier* ($M$ = 4.43) is just above the scale midpoint, suggesting that participants perceive eating context as a potential barrier to using personalised nutrition advice. Finally, the mean score on *Intention to Use Personalised Nutrition Advice* ($M$ = 3.86), which is just below the scale midpoint, suggests only a moderate intention to make use of personalised recommendations.

In addition, Table 3 shows that *Intention to Use Personalised Nutrition Advice* is moderately to strongly correlated with all variables except *Eating Context Barrier*, which is not significantly correlated, and *Privacy Risk*, which is only weakly correlated [48]. However, the moderately positive correlation between *Eating Context Barrier* and *Ambivalent Feelings* suggests that the perceived barriers of the eating context could have an indirect negative effect on the intention to adopt personalised nutrition services, as the result of an increase in ambivalent feelings.

## 3.2 Measurement model

Based on the model fit criteria, the results indicated a satisfactory fit of the measurement constructs to the underlying data: $\chi^2$/df = 303.5/121 = 2.51, RMSEA = 0.044, SRMR = 0.041, CFI = 0.98, and TLI = 0.98. In addition, as can be seen in Table 2, the variables and their items had high factor loadings, high composite reliabilities (Cronbach's alphas >0.70), and a positive average variance extracted (i.e., AVE >0.50).

## 3.3 Structural models estimation

In order to estimate the structural models, we added the regression paths one by one. First, a model was estimated that included the association of *Personalisation Benefit* and *Privacy Risk* with *Intention to Use Personalised Nutrition Advice*, with *Risk-Benefit Calculus* as mediator. The results showed a relatively poor model fit ($\chi^2$/df = 5.69; RMSEA = 0.077; SRMR = 0.083; CFI = 0.974; TLI = 0.964). This model confirmed previous studies suggesting that the more positive the outcome of the *Risk-Benefit Calculus*, the higher the participants' *Intention to Use Personalised Nutrition Advice* (b = .67; $p < .001$). Additionally, *Personalisation Benefit* had a positive effect on the outcome of the *Risk-Benefit Calculus* (b = .62; $p < .001$), whereas *Privacy Risk* had a negative effect on the outcome of the *Risk-Benefit Calculus* (b = -.20; $p < .001$). Note that the effect of *Personalisation Benefit* was much stronger than the effect of *Privacy Risk*.

Our objective was to examine whether including *Ambivalent Feelings* as an additional mediator next to *Risk-Benefit Calculus* significantly improved the model fit. Therefore, we estimated a model with *Ambivalent Feelings* as an additional mediator. The results indicated an improved model fit, although still not meeting all criteria ($\chi^2$/df = 5.00; RMSEA = 0.071; SRMR = 0.073; CFI = 0.969; TLI = 0.959) (see Table 4 for a comparison of the different models). The model indicated that next to *Risk-Benefit Calculus* (b = .58; $p < .001$), *Ambivalent Feelings* was a significant, negative predictor of *Intention to Use Personalised Nutrition Advice* (b = -.15; $p < .001$), suggesting that the intention to use personalised nutrition advice increases when the 'benefits and risks' trade-off is more positive, but decreases when participants experience more ambivalent feelings. Furthermore, *Personalisation Benefit* was not only related

**Table 4. Model comparison.**

| | Fit indices | | | | | | |
|---|---|---|---|---|---|---|---|
| | Chi-square | df | CFI | TLI | RMSEA | SRMR | $\Delta\chi^2$ ($\Delta$df) |
| Model I: *Personalisation Benefit, Privacy Risk as independent variables, and Risk-Benefit Calculus as mediator* | 182.06 | 32 | .974 | .964 | .077 | .083 | |
| Model II: *Ambivalence as an additional mediator* | 294.98 | 59 | .969 | .959 | .071 | .073 | 112.9 (27)** |
| Model III: *Eating Context Barrier as additional independent variable* | 378.64 | 124 | .972 | .966 | .051 | .058 | 83.7 (65)** |

df = degrees of freedom, CFI = Comparative Fit Indices, TLI = Tucker-Lewis Indices, RMSEA = Root Mean Square Error, SRMR = Standardized Root Mean Square Residual, $\Delta\chi^2$ = Chi-square difference, $\Delta$df = difference in degrees of freedom;

* $p < .05$;

** $p < .01$.

positively to *Risk-Benefit Calculus* (b = .61; $p < .001$), but also negatively to *Ambivalent Feelings* (b = -.39; $p < .001$), suggesting that participants feel less ambivalent when more personalisation benefits are perceived. In a similar vein, *Privacy Risk* was not only related negatively to *Risk-Benefit Calculus* (b = -.20; $p < .001$), but also positively to *Ambivalent Feelings* (b = .45; $p < .001$), suggesting that participants experience more ambivalent feelings when more privacy risks are perceived. In contrast to the effect on *Risk-Benefit Calculus*, the effect of *Privacy Risk* on *Ambivalent Feelings* was stronger than the effect of *Personalisation Benefit*. In other words, perceived privacy risk (in contrast to perceived personalisation benefits) is more strongly related to the perception of ambivalent feelings, whereas the perception of personalisation benefits (in contrast to perceived privacy risks) is more strongly related to the trade-off between benefits and risks.

As a final step, the model was further extended with *Eating Context Barrier* as an additional independent variable. In contrast to the previous two models, this model indicated a good model fit ($\chi^2$/df = 3.05; RMSEA = 0.051; SRMR = 0.058; CFI = 0.972; TLI = 0.966). Similar to the previous model, *Risk-Benefit Calculus* (b = .57; $p < .001$) was positively related and *Ambivalent Feelings* (b = -.20; $p < .001$) was negatively related to *Intention to Use Personalised Nutrition Advice*. These effects suggest that intention to use personalised nutrition advice is higher when more benefits than risks are perceived and lower when more ambivalent feelings are experienced. *Personalisation Benefit* was still positively related to *Risk-Benefit Calculus* (b = .62; $p < .001$) and negatively related to *Ambivalent Feelings* (b = -.43; $p < .001$) and *Privacy Risk* was negatively related to *Risk-Benefit Calculus* (b = -.19; $p < .001$) and positively related to *Ambivalent Feelings* (b = .37; $p < .001$). In addition, *Eating Context Barrier* in this model was positively related to *Ambivalent Feelings* (b = .39; $p < .001$) but not significantly related to *Risk-Benefit Calculus* (b = -.06; $p = 0.155$). This suggests that the more eating context is perceived as a barrier to using personalised nutrition advice, the more ambivalent feelings one experiences. Additionally, *Eating Context Barrier* suggested a positive direct relationship with *Intention to Use Personalised Nutrition Advice* (b = .22; $p < .001$), suggesting that intention to use personalised nutrition advice increases when the eating context is perceived more as a barrier. See Fig 1 for the final model.

## 4. Discussion

### 4.1 General discussion

This study developed and tested a model in which we looked at whether ambivalent feelings and perceived eating context barriers may further explain consumers' intentions to use

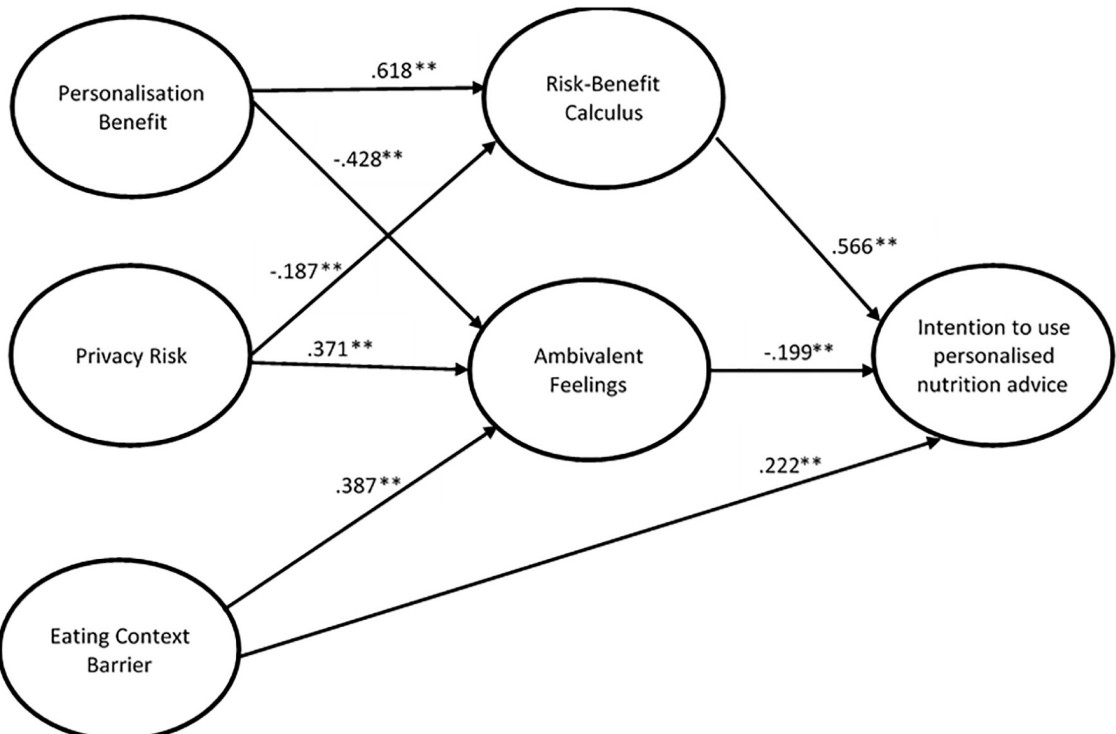

**Fig 1. Final model–Significant paths.** Note: Only the significant path coefficients of the latent variables are shown. All item loadings were significant. For reasons of clarity, we decided not to report the item loadings and standardized error variances. * p <.05; ** p <.01.

personalised nutrition services above the perceived benefits of receiving personalised nutrition advice and the perceived privacy risks. More specifically, this study provides insight into how these factors relate to the trade-off between perceived risks and benefits (as captured by the concept of risk-benefit calculus) and intention to adopt personalised nutrition services. We will discuss the findings of the current study in more detail below.

First, the current study (partly) replicated previous research by examining the relationships between perceived personalisation benefits, perceived privacy risks, and the intention to make use of personalised nutrition advice. The strong positive correlation found between risk-benefit calculus and the intention to use personalised nutrition advice, with risk-benefit calculus being the result of weighing benefits and (privacy) risks, is indeed in agreement with previous research. This finding suggests that the intention to make use of personalised nutrition advice is greater when the risk-benefit calculus is evaluated more positively (i.e. more benefits than risks are perceived). Additionally, participants tend to agree more with the benefits of personalised nutrition than with its associated privacy risks—that is, the benefits of personalised nutrition have a stronger effect than the risks of privacy loss on the intention to use personalised nutrition services through risk-benefit calculus. This finding is in line with previous findings from the literature. Both Poínhos et al. and Berezowska et al. found that perceived benefits had a stronger influence on the intention to adopt personalised nutrition advice than perceived privacy risks [19,22].

Second, this study extends previous research by suggesting the additional value of ambivalent feelings as an affective construct in predicting intention to use personalised nutrition services. Stated differently, feelings of conflict, uneasiness and mixed feelings evoked by the

concept of personalised nutrition are negatively related to the usage intention of these types of advice. More specifically, this study indicates a negative association of ambivalent feelings with intentions, beyond the cognitive weighing of perceived benefits and risks by means of the risk-benefit calculus. These findings suggest that conflicting evaluations of personalisation benefits and privacy risks may result in a hesitancy regarding the concept of personalised nutrition (e.g. is it beneficial or risky?), which decreases one's adoption intention. Furthermore, these findings are in line with a relatively novel body of studies in which ambivalent feelings are also highly relevant in shaping consumers' intentions towards novel products or concepts [49]. According to these studies ambivalence may be experienced as an emotional state, which may be used as information regarding the specific object [50], which is in our case personalised nutrition. This observation also corresponds with the feeling-as-information theory, which argues that individuals use their feelings as a source of information in behavioural responses [51].

Finally, and in addition to the effect of ambivalent feelings, this study extends previous research by including eating context as a possible barrier to the adoption of personalised nutrition services. Besides the explorative survey by Stewart-Knox et al. [33], few studies on personalised nutrition have investigated the role of an individual's eating context as it relates to the acceptance of personalised nutrition advice. More specifically, this study suggests that the perception of the eating context as a barrier to the use of personalised nutrition advice positively contributes to ambivalent feelings towards the adoption of personalised nutrition. This suggests that an individual's eating context may indeed act as a barrier to the usage of personalised nutrition services, albeit indirectly, primarily through enhanced ambivalent feelings. Strikingly, this study indicated that, in addition to this indirect effect through ambivalent feelings, eating context barriers also positively affect the intention to use personalised nutrition advice. A possible explanation for the positive correlation between eating context and intention could be that those who intend to adopt personalised nutrition, may have more thoroughly considered the logistics involved, making them more aware of the potential social and contextual barriers to compliance with an individualised diet. The positive correlation of eating context barriers might also be due to measurement error; the observed correlation may be an indirect effect induced by the other predictor variables. We expect this with greater reason, because eating context barriers show no significant inter-correlation with intention to use personalised nutrition advice, while they do positively affect ambivalent feelings, which in turn negatively affect intention to use personalised nutrition advice. More research is needed to further elucidate why these perceived eating context barriers may positively affect the adoption intention of personalised nutrition services. Nevertheless, the implication would be that personalised advice should consider the individual's eating context.

## 4.2 Limitations and future research

A number of limitations exist in this study. First and foremost is that we only focused on a limited number of constructs that may play a role in consumers' evaluation and usage intention of personalised nutrition services. There may be other factors that play a role in the evaluation of the concept of personalised nutrition that were not considered in this study, such as trust in different types of providers of these services [52,17], one's personal motivation and ability to change behaviour [53], and the way or format with which personalised nutrition advice is communicated to consumers [54]. Similarly, the perceived risks that we included in this study are limited to data and privacy issues. It could very well be that consumers see other risks, that have not been included in this study. Equally, while only eating context barriers are included as possible barriers, consumers could perceive other barriers in the adoption of personalised nutrition services. For example, consumers may find it too effortful or complex to conduct do-

it-yourself health measurements and filling in questionnaires or food diaries in order to receive personalised advice. Future research could help to identify other risks or barriers that consumers may encounter when adopting personalised nutrition services.

In addition, while ambivalent feelings have been included in the model, other positive or negative emotions were not. Onwezen et al. discuss the motivational function of ambivalence via emotions and argue that it remains unclear to what extent the effect of ambivalence comes from mixed feelings, or from positive versus negative emotions [49]. The authors conclude that, while ambivalence is a concept that differs from (the weighing of) perceived risks and benefits, it is also different from emotions. We did not control for the effect of emotions in this study, and can therefore not provide clarity in that respect. Future research could include positive and negative emotions as additional independent variables to potentially increase the explained variance and to further unravel the role played by ambivalent feelings in one's intention to adopt personalised nutrition services.

A major limitation of this study is that only one definition of personalised nutrition services is used in the research: personalised nutrition is based on personal preferences and/or self-formulated goals and/or eating habits and/or health status and/or DNA. This description of personalised nutrition is very general; participants may have reacted differently to different examples of how personalised nutrition may look like. Consequently, it is possible that the perceived benefits and risks of personalised nutrition services may be different for those individuals who considered personalised nutrition services based on DNA as opposed to individuals who considered personalised nutrition services that only take into account everyday eating habits. Furthermore, the examples that were provided to illustrate the definition could be perceived by respondents as too generic and not personally relevant. This may have led consumers to view the concept of personalised nutrition as a gimmick. Since our study did not control whether participants may have felt this way, our findings need to be interpreted carefully. Future research may refine the insights obtained in the current study by including a variety of more specific types of (fictional) nutritional advice to further examine how individuals react to different types of personalised nutrition recommendations. Additionally, to further reduce ambiguity in the findings, it could be helpful to guide participants, for example in the 'Introduction' part of the questionnaire, to answer the questions based on a concrete situation where the use of personalised nutrition may be relevant.

Another, related potential limitation that may be associated with this study is that self-reported measures were used to identify consumers' intention to use personalised nutrition services. Self-reported questionnaires have some known disadvantages. For example, respondents may not answer truthfully (social desirability bias) or may show a tendency to respond in a certain way regardless of the question (response bias). To overcome these biases, and to further test the consistency of the used measurement scales (i.e., how individuals are answering to the used measures at any given time), future research may consider a longitudinal data collection, which may also be used to assess test-retest reliability. Moreover, because the concept of personalised nutrition is a new concept, response quality may have been affected by a lack of direct experience with personalised nutrition. While a definition was provided at the beginning of the survey to familiarise respondents with personalised nutrition, the definition is, as previously mentioned, still rather generic and abstract. This may have made it difficult for participants to clearly envision the benefits and risks of personalised nutrition.

An additional limitation could relate to the generalisability of the results to other countries. The sample that has been used is representative of the adult Dutch population in terms of gender, age, level of education and income. Future research should replicate the study in other countries to judge the generalisability of the findings. Nevertheless, the observation that our findings are in line with Berezowska et al. and Stewart-Knox et al. [22,33], who included eight

or nine European countries in their studies, makes us confident that our findings are at least typical of other European countries. Berezowska et al. state that their cognitive model, which includes personalisation benefits, perceived privacy risks and the privacy calculus is robust and applicable throughout Europe [22]. Stewart-Knox et al. state that the Netherlands rated eating context as a significantly lower barrier than did other European countries [33], which may suggest that the importance of eating context will be even more important when our model is tested in other European countries.

Future research may further investigate the relation between eating context and intentions to adopt personalised nutrition services by, for example, further specifying the different contextual factors that may affect an individual's evaluation of personalised nutrition services. Moreover, future research could investigate how the enhancing effect of eating context on ambivalent feelings can be reduced, perhaps, by including contextual factors such as preferences or dietary requirements of family members in the personalised nutrition advice.

Finally, and in consideration of the previous point, it would be interesting in future research to distinguish between different types of consumers or consumer groups. In this study we developed a generic model, but it is possible that the relations between the different concepts will work differently with diverse groups. For example, future research could distinguish between the degree of experience that various people have with the concept of personalised nutrition, meaning that different interventions, services or communications could be effective in varying degrees for different consumer groups. It would be beneficial for future research to explore how our results can be used effectively in real-life implementations of personalised nutrition services and whether it would be effective to differentiate between several consumer groups.

## 5. Conclusions and practical implications

The current study suggests the importance of considering ambivalent feelings and eating context barriers in understanding consumers' intentions to adopt personalised nutrition services. This research has practical relevance, as its findings may potentially be used to design personalised nutrition services that address the specific needs and concerns of consumers. Based on the findings of the current study, we offer the following practical implications for stimulating the adoption of personalised nutrition services.

First, this study indicates that it is essential to communicate clearly the benefits of personalised nutrition, which may differ among individuals (e.g. losing weight, gaining weight, improving health). It is important that these benefits appeal not only to a small group of motivated, often highly educated consumers who have already established a healthy lifestyle, but should also take into account the preferences and needs of broader societal groups. Note in this respect that Adams et al. [3] argue that '*if personalised nutrition is effective in sustaining changes in health, then it is desirable that these tools are accessible and affordable to most people*' (p. 4).

Additionally, personalised nutrition services should take privacy issues seriously and communicate transparently about how personal data are protected. Similarly, providers of personalised nutrition should disclose their intentions for customer data use. For example, individuals who may be interested in participating in a personalised nutrition program should be informed that their profiles will not be shared with third parties who may use this information for commercial reasons. Security and anonymity of data should be made transparent to customers of personalised nutrition services, who should have the opportunity to provide informed consent. Additionally, an individual's data or measures should only be collected if there is reliable evidence that personalising advice based on this information could influence health or deliver functional benefits [3]. There should also be transparent communication as to how each piece of the collected information contributes to these (health) benefits.

Personalised nutrition services should address and try to reduce ambivalent feelings. This implies that it is important to provide clear benefits and tackle privacy risks, and also suggests the importance of the way in which these aspects are communicated. In their communication, personalised nutrition service providers should take seriously consumers' feelings and concerns. Furthermore, communicating in a clear and understandable way and employing user-friendly tools could help to further reduce ambivalence.

Finally, individuals' eating context should be taken into account by, for example, facilitating personalised dietary advice in the workplace and in the social environments where people eat. In that respect, personalised nutrition advice would benefit from considering the eating preferences of others in one's eating context in addition to the preferences of the individual for whom the advice was designed. Service providers that make use of apps with location-based services, for example by using GPS technology, can take into account the different locations where their customers eat, adjusting their services and recommendations accordingly. Alternatively, users of personalised services could indicate their preferences themselves given certain eating contexts, which could then be implemented into personalised recommendation systems based on underlying algorithms.

## Supporting information

**S1 Data. Original questionnaire.** Original questionnaire (in Dutch).
(PDF)

**S2 Data. Translated questionnaire.** Translated questionnaire (in English).
(PDF)

**S3 Data. SPSS raw datafile.**
(SAV)

## Author Contributions

**Conceptualization:** Machiel J. Reinders, Emily P. Bouwman, Jos van den Puttelaar, Muriel C. D. Verain.

**Formal analysis:** Machiel J. Reinders, Muriel C. D. Verain.

**Methodology:** Machiel J. Reinders.

**Writing – original draft:** Machiel J. Reinders.

**Writing – review & editing:** Machiel J. Reinders, Emily P. Bouwman, Jos van den Puttelaar, Muriel C. D. Verain.

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
