## [Decision Letter · Decision Letter 0]

27 Nov 2019

PONE-D-19-26890

Consumer acceptance of personalised nutrition: The role of ambivalent feelings and eating context

PLOS ONE

Dear Dr Reinders,

Thank you for submitting your manuscript to PLOS ONE. After careful consideration, we feel that it has merit but does not fully meet PLOS ONE’s publication criteria as it currently stands. Therefore, we invite you to submit a revised version of the manuscript that addresses the points raised during the review process.

Below and attached you will find two comprehensive peer reviews conducted on your manuscript. As neither reviewer was comfortable checking your statistical analysis I will ask that a statistician also review while you address the reviewer comments.

We would appreciate receiving your revised manuscript by Jan 11 2020 11:59PM. To enhance the reproducibility of your results, we recommend that if applicable you deposit your laboratory protocols in protocols.io, where a protocol can be assigned its own identifier (DOI) such that it can be cited independently in the future. For instructions see: http://journals.plos.org/plosone/s/submission-guidelines#loc-laboratory-protocols

We look forward to receiving your revised manuscript.

Kind regards,

Katie MacLure, PhD, MSc (dist)., BSc (1st)

Academic Editor

PLOS ONE

Journal Requirements:

3.  Please include additional information regarding the survey or questionnaire used in the study and ensure that you have provided sufficient details that others could replicate the analyses. For instance, if you developed a questionnaire as part of this study and it is not under a copyright more restrictive than CC-BY, please include a copy, in both the original language and English, as Supporting Information. Moreover, please include more details on how the questionnaire was pre-tested, and whether it was validated.

4.  We noticed you have some minor occurrence of overlapping text with your following previous publication(s), which needs to be addressed:

- Onwezen, Marleen C., Machiel J. Reinders, and Siet J. Sijtsema. "Understanding intentions to purchase bio-based products: The role of subjective ambivalence." Journal of Environmental Psychology 52 (2017): 26-36.

The text that needs to be addressed involves some sentences of the Discussion.

In your revision ensure you cite all your sources (including your own works), and quote or rephrase any duplicated text outside the methods section. Further consideration is dependent on these concerns being addressed.

Reviewers' comments:

Reviewer's Responses to Questions

**Comments to the Author**

1. Is the manuscript technically sound, and do the data support the conclusions?

Reviewer #1: Yes

Reviewer #2: Partly

2. Has the statistical analysis been performed appropriately and rigorously? 

Reviewer #1: I Don't Know

Reviewer #2: I Don't Know

3. Have the authors made all data underlying the findings in their manuscript fully available?

Reviewer #1: Yes

Reviewer #2: No

4. Is the manuscript presented in an intelligible fashion and written in standard English?

Reviewer #1: Yes

Reviewer #2: No

5. Review Comments to the Author

Reviewer #1: The paper explores factors that could affect the uptake of personalised nutrition advice, and contributes to our understanding of barriers to behaviour change following tailored nutrition advice. The authors have provided a detailed background to their research and discussed their results in the context of the literature.

I do not have the expertise to comment on the statistical approach taken, but it does seem to be comprehensively described.

My main comments are with regard to English and grammar – noted below. Otherwise, the manuscript is mainly very well written and the suggested revisions are minor.

• Abstract, line 12 – check use of apostrophe for ‘individuals’

• Abstract, line 22 – I think ‘inversely’ might be a more appropriate word than ‘reversely’.

• Abstract, line 25 - ‘In conclusion’ would be better than ‘In sum’.

• Introduction, line 42 – I cannot see Brug et al (1999) in the reference list.

• Introduction, line 43 – I think the authors need a reference to support the statement that more personally relevant nutrition information will lead to higher compliance. Otherwise, they could state ‘and may in turn lead to greater compliance’.

• Introduction, line 45 – check use of apostrophe in ‘individuals’

• Introduction, line 50 – is a reference needed to support the statement that there is a high penetration rate of smartphone ownership? Also, perhaps ‘high prevalence’ is a better phrase than ‘high penetration rate’?

• Introduction, line 52 – check sentence structure. I think it needs to read ‘(e.g. smart wearables) are becoming rapidly available’.

• Introduction, line 53 - a reference is needed to support the statement that consumer interest in these tools is growing.

• Introduction, line 99 – check use of apostrophe in ‘consumers’.

• Introduction, line 110 – ‘a’ is not needed, i.e. ‘adhering to personalised advice’.

• Introduction, line 111 – ‘advice’ is the plural.

• Introduction, line 111 – no comma needed after ‘preferences’.

• Introduction, line 115 - check use of apostrophe in ‘consumers’.

• Introduction, lines 118-122 – I think this is better in the Discussion, and the Introduction should end with the aim of the study.

• Methods, line 130 – ‘voluntariness’ is not a word.

• Methods, lines 146-147 – please explain what is meant by low, moderate and high education level.

• Table 1 – please give SD for age as well as the range.

• Methods, line 194 – please explain why means and SD are presented, yet Spearman’s rank correlation was performed. It is more usual to see medians presented if using Spearman’s correlations.

• Table 4 – it would be useful for abbreviations to be explained in footnotes to the table.

• Discussion – please discuss the generalisability of your findings, i.e. are the consumer panels considered representative of the wider population?

• Discussion, line 315 – please check use of apostrophe in ‘consumers’ – consumers’?

• Discussion, line 368 – ‘advices’ is incorrect. Suggest rephrase to ‘(fictional) examples of nutritional advice’.

Reviewer #2: 1. Partly: I feel that the authors need to include the questionnaire in their paper in exactly the way it was presented to their participants. The findings are confusing and ambiguous without having the questionnaire included.

2. I don't know: The authors did not state if they carried out a 'normalcy distribution' test. I cannot determine if how they chose to represent their data is actually the best way to represent it. They did not justify why they are using the mean and standard deviation.

3. No: this is connected with the first question, where I feel it is highly relevant to include the exact questionnaire that was used for their study.

4. No: The manuscript needs more work. Specifically, the methods, discussion and conclusion need to be written more scientifically, and colloquialisms need to be removed.

6. PLOS authors have the option to publish the peer review history of their article (what does this mean?). If published, this will include your full peer review and any attached files.

Reviewer #1: No

Reviewer #2: Yes: Dr Rachael H Sibson

---

## [Author Response · Author response to Decision Letter 0]

10 Jan 2020

A rebuttal letter that responds to each point raised by the academic editor and reviewer(s) is uploaded as separate file and labeled 'Response to Reviewers'.

---

## [Decision Letter · Decision Letter 1]

4 Feb 2020

PONE-D-19-26890R1

Consumer acceptance of personalised nutrition: The role of ambivalent feelings and eating context

PLOS ONE

Dear Dr Reinders,

Thank you for submitting your revised manuscript to PLOS ONE. The two original reviewers looked over your revisions and thanked you for addressing all their points. However, they have both noted further minor revisions which would strengthen your article. Therefore, we invite you to submit a further revised version of the manuscript.

We would appreciate receiving your revised manuscript by Mar 20 2020 11:59PM. To enhance the reproducibility of your results, we recommend that if applicable you deposit your laboratory protocols in protocols.io, where a protocol can be assigned its own identifier (DOI) such that it can be cited independently in the future. For instructions see: http://journals.plos.org/plosone/s/submission-guidelines#loc-laboratory-protocols

We look forward to receiving your revised manuscript.

Kind regards,

Katie MacLure, PhD, MSc (dist)., BSc (1st)

Academic Editor

PLOS ONE

Reviewers' comments:

Reviewer's Responses to Questions

**Comments to the Author**

1. If the authors have adequately addressed your comments raised in a previous round of review and you feel that this manuscript is now acceptable for publication, you may indicate that here to bypass the “Comments to the Author” section, enter your conflict of interest statement in the “Confidential to Editor” section, and submit your "Accept" recommendation.

Reviewer #1: All comments have been addressed

Reviewer #2: All comments have been addressed

2. Is the manuscript technically sound, and do the data support the conclusions?

Reviewer #1: Yes

Reviewer #2: Yes

3. Has the statistical analysis been performed appropriately and rigorously? 

Reviewer #1: I Don't Know

Reviewer #2: I Don't Know

4. Have the authors made all data underlying the findings in their manuscript fully available?

Reviewer #1: Yes

Reviewer #2: Yes

5. Is the manuscript presented in an intelligible fashion and written in standard English?

Reviewer #1: Yes

Reviewer #2: Yes

6. Review Comments to the Author

Reviewer #1: The authors should be commended on the care taken to address all the comments. I am happy that my comments have been addressed. Whilst there is some duplication of means and SD between tables 2 and 3, I can see why the authors have included this. On reading the paper, I did notice a few minor corrections required to English:

• Line 73 – should be ‘received’ instead of ‘receiving’

• Line 124 – should be ‘To ensure a sample that was nationally representative’ instead of ‘is’.

• Line 161 – should read ‘Before data collection started’.

• Line 376 – should read ‘suggests that’ instead of ‘suggest that’

• Line 433 – should be ‘advice’ instead of ‘advices’.

Reviewer #2: Q1. Yes, the authors addressed each issue raised by me.

Q2. To the best of my knowledge, the manuscript is acceptable, however, I think one issue still needs to be addressed, and that is, precisely who were the individuals this study was targeting, or trying to get information from? Also, a minor issue is if the questionnaire was validated.

Q3. I do not know enough about the statistics used here to appropriately comment on.

Q4. I believe the answer is Yes.

Q5. The revised manuscript is much more intelligible.

7. PLOS authors have the option to publish the peer review history of their article (what does this mean?). If published, this will include your full peer review and any attached files.

Reviewer #1: No

Reviewer #2: Yes: Dr Rachael H Sibson

---

## [Author Response · Author response to Decision Letter 1]

6 Mar 2020

We addressed the comments of the editor and reviewers in a rebuttal letter, which is uploaded as separate file and labeled 'Response to Reviewers'.

---

## [Editor Report · Decision Letter 2]

23 Mar 2020

Consumer acceptance of personalised nutrition: The role of ambivalent feelings and eating context

PONE-D-19-26890R2

Dear Dr. Reinders,

We are pleased to inform you that your manuscript has been judged scientifically suitable for publication and will be formally accepted for publication once it complies with all outstanding technical requirements.

With kind regards,

Katie MacLure, PhD, MSc (dist)., BSc (1st)

Academic Editor

PLOS ONE
---

## [Editor Report · Acceptance letter]

27 Mar 2020

PONE-D-19-26890R2 

Consumer acceptance of personalised nutrition: The role of ambivalent feelings and eating context 

Dear Dr. Reinders:

I am pleased to inform you that your manuscript has been deemed suitable for publication in PLOS ONE. Congratulations! Your manuscript is now with our production department. 

With kind regards,

on behalf of

Dr. Katie MacLure 

Academic Editor

PLOS ONE